# Novel Soft Haptic Biofeedback—Pilot Study on Postural Balance and Proprioception [note 1]

**DOI:** 10.3390/s22103779

**Published:** 2022-05-16

**Authors:** Mert Aydin, Rahim Mutlu, Dilpreet Singh, Emre Sariyildiz, Robyn Coman, Elizabeth Mayland, Jonathan Shemmell, Winson Lee

**Affiliations:** 1Applied Mechatronics and Biomedical Engineering Research (AMBER) at School of Mechanical, Materials, Mechatronic and Biomedical Engineering, University of Wollongong, Wollongong, NSW 2522, Australia; ma731@uowmail.edu.au (M.A.); emre@uow.edu.au (E.S.); ccwlee@uow.edu.au (W.L.); 2Intelligent Robotics & Autonomous Systems Co (iR@SC), RA Engineering, Shellharbour, NSW 2529, Australia; 3Biofabrication and Tissue Morphology (BTM) Group, Centre for Biomedical Technologies, Faculty of Engineering, Queensland University of Technology, Brisbane, QLD 4000, Australia; dilpreetsingh8@hdr.qut.edu.au; 4School of Health & Society, University of Wollongong, Wollongong, NSW 2522, Australia; rcoman@uow.edu.au; 5School of Health Sciences, Western Sydney University, Sydney, NSW 2560, Australia; b.mayland@westernsydney.edu.au; 6School of Medical, Indigenous and Health Sciences, University of Wollongong, Wollongong, NSW 2522, Australia; shemmell@uow.edu.au

**Keywords:** soft haptic, biofeedback, proprioception, soft sensor, 3D printing

## Abstract

Sensory feedback is critical in proprioception and balance to orchestrate muscles to perform targeted motion(s). Biofeedback plays a significant role in substituting such sensory data when sensory functions of an individual are reduced or lost such as neurological disorders including stroke causing loss of sensory and motor functions requires compensation of both motor and sensory functions. Biofeedback substitution can be in the form of several means: mechanical, electrical, chemical and/or combination. This study proposes a soft monolithic haptic biofeedback device prototyped and pilot tests were conducted with healthy participants that balance and proprioception of the wearer were improved with applied mechanical stimuli on the lower limb(s). The soft monolithic haptic biofeedback device has been developed and manufactured using fused deposition modelling (FDM) that employs soft and flexible materials with low elastic moduli. Experimental results of the pilot tests show that the soft haptic device can effectively improve the balance of the wearer as much as can provide substitute proprioceptive feedback which are critical elements in robotic rehabilitation.

## 1. Introduction

Sensory feedback is critical in proprioception to orchestrate muscles to perform targeted motion(s). Sensory feedback gains even further significance in the rehabilitation of neurological disorders such as stroke as a worldwide health problem requiring compensation of motor and sensory functions which may be augmented by biofeedback devices [1]. Brain regions devoted to sensory and motor functions are affected during stroke resulting in the loss of many neurons critical for feedback movement control. This results in reducing a person’s sense of their limbs and their capacity for muscular force production along with spasticity, an increase in the resistance of a muscle(s) to lengthening [2,3]. Neurological rehabilitation largely involves extensive physio- and occupational therapy interventions that aim to re-establish functional sensory and motor pathways in the brain, thereby restoring mobility, functional use of limbs, and reducing impairment and disability [4].

Biofeedback plays a significant clinical role in the rehabilitation process of patients with reduced sensory feedback [5]. Various feedback methods are used in the process such as auditory, visual and tactile systems, and combination(s) of those systems. These feedback means can improve the sensing ability of the patients about the kinematics, kinetics and muscular activities of their affected body parts [5] by compensating for the loss of sensory information previously delivered through the proprioceptive system. Multiple studies suggest that neurological rehabilitation such as stroke should consist of intensive and multisensory task-related training [6,7]. Various forms of haptic feedback, including mechanical, electrical and vibrational systems, have been demonstrated to enhance the utility of sensory feedback for patients with proprioceptive dysfunction or loss [8]. Many of the haptic feedback studies examine the use of vibrotactile stimulation by employing commercially available, miniature and lightweight vibration motors that can easily and directly be placed onto the skin of a patient [9]. Although these mechatronic systems are advocated as non-invasive and low-cost feedback systems due to the fact that the components used in these systems are easily-accessible in the commercial market, their development is complicated, and they require complex and sophisticated control algorithms.

The field of soft robotics has gained a great amount of interest due to the numerous advantages it has in comparison to the conventional robotics field. This rapidly growing research field combines expertise from various disciplines including materials science, mechanical engineering, electrical engineering, control engineering, chemistry, physics, computer science, biology, medicine and many others [10]. Soft robotic systems have many interesting advantages such as adaptability, conformability, agility, and high compliance [11,12]. In addition, these systems can operate for very long periods of time without failure since they are made of highly elastic and deformable materials which, in general, operate under the rate of stress levels in the material [13]. More importantly, soft robotic systems can safely interact with humans and operate in highly dynamic environments [14]. The realization of robotic systems with high biomimicry remains the focus of soft roboticists. One way to achieve this aim is to fully exploit the material properties involved in the fabrication of soft robotic devices. Primarily, the use of materials with low elastic moduli has effectively led soft robots to mimic their natural counterparts with hyper-deformations and minimal structural stress [15]. Conventional robotic systems are made of stiff and rigid components that are ideal for precise and repetitive tasks that require high forces and fast movements, however, their safety is questionable to operate alongside humans due to their rigidity, speed of operation, and thus physical impact in case of any collision with a human. In addition, the manufacturing and development of such systems require complex machining, laborious assembly processes and the incorporation of multiple physical sensors [16].

Among various examples of soft robots with prime inspiration from the movements of their natural counterparts, soft robotic systems can serve as biofeedback devices that can non-invasively be placed on the human body communicating useful information to substitute and/or augment our sensory system. Soft, wearable and pneumatically actuated haptic feedback systems in this matter have great potential as haptic feedback devices since they are great candidates for a safe human-machine/computer interaction due to their softness and compliance [10,17]. Abd et al. demonstrated a soft haptic armband that can be used as a feedback system when operating with a robotic hand [18]. The tactile data obtained from the robotic hand was delivered back to the user to enhance the teleoperation experience. The soft haptic armband was fabricated using silicone rubber that was moulded and casted using conventional techniques to produce soft pneumatic chambers that inflate when activated. Zhao et al. developed a soft haptic communicator with dielectric elastomer actuators, which is capable of delivering indentation force as low as 0.6 N against the skin of a wearer’s arm at the lowest operating frequency [19]. We have developed an alternative soft haptic biofeedback device integrated with force sensing capability that employs the same actuation as the soft haptic device used by Zhao et al. to assess its efficacy for inducing biofeedback of the wearer [20]. Our soft haptic device is actuated pneumatically while sensing pressure applied onto the skin of the wearer from the fully 3D printed soft force sensor operating as a resistive sensor.

Conventional manufacturing methods limit the potential use of soft robotic devices due to imposed constraints such as geometry, size and shape on pneumatic actuators. These limitations can be alleviated using additive manufacturing techniques (i.e., 3D printing) to produce soft robots with complex shapes and geometries based on computer-aided design (CAD) models of actuation [21] and sensing systems [22]. In this study, we prototyped a soft haptic biofeedback device comprising three units of soft bellow actuators which were fully 3D printed using a low-cost and open-source fused deposition modelling (FDM) 3D printer using a commercially available soft and flexible thermoplastic poly(urethane) (TPU). The soft bellow pneumatic actuator that generates a linear stroke upon activation was designed and implemented as the actuation unit. Modular compliant tips on the head of the unit were designed with various shapes to assess the comfort and sensation levels of subjects when the bellow actuator is activated using positive pressure to push the tip of the unit against the skin of the user. Among several modular compliant tips, the pointy tip, as shown in Figure 1 was found to be the most effective for stimulation and biofeedback while more round and flat tips distribute pressure, thus losing their effectiveness. The CAD model and the fabricated soft haptic biofeedback device proposed and the unit proposed in this study are shown in Figure 2.

The major contribution of this study is prototyping a soft biofeedback device utilizing multiple monolithic and soft haptic units integrated with a soft Velcro strap that can be directly fabricated using a low-cost 3D printing technology. The soft haptic biofeedback device presented in this work can be a benefit for various robotic applications such as human–robot/computer-interactions. Robot-assisted rehabilitation is also one of the areas that requires biofeedback device(s) to step towards a more comprehensive rehabilitation compansating motor and sensory function loss. The soft haptic biofeedback device prototyped in this study as shown in Figure 2 is quite compact in size, and it can generate high force outputs with great potential to augment proprioception in human–robot/computer interactions and/or compensate for reduced proprioception in neurological rehabilitation. In this study, we propose a soft haptic biofeedback system that comprises of the three soft haptic units which were 3D printed from TPU material using the FDM type additive manufacturing. Soft bellow type soft actuators have been designed with minimal wall thickness. Biofeedback is provided by the device inflating and pressing onto the skin of the wearer’s lower limb, activating cutaneous receptors in the skin. The soft haptic biofeedback units have modular tips with various shapes that enable the comfort and sensation levels of subjects to be assessed and optimized. This study conducts a pilot investigation on the effectiveness of the soft haptic biofeedback device with participating healthy subjects for their balance and proprioception and reports the device performance in these pilot tests whereas experiments with a larger group of subjects will be conducted with a wider spectrum of participant criteria including healthy as much as subjects with a loss/reduced sensory function by administering further safety protocols in our future studies. The soft haptic biofeedback system was operated with a centre of pressure (CoP) measurement device equipped with force sensitive resistors (FSR) as force sensing components in order to obtain the CoP information of the subject which will help to evaluate the subject’s posture and balance information. The pilot study also investigates the proprioception of the subject upon postural balance tests enquiring the subject to recall their body position in different conditions.

## 2. Soft Haptic System with Soft Force Sensor

Two soft and flexible commercially available materials were used to fabricate the soft haptic unit. The first material which is a TPU known as NinjaFlex (NinjaTek, Manheim, PA, USA) was used to 3D print the soft pneumatic bellow and the soft sensor base and tip as shown in Figure 1. The second material, which is a conductive black filament based on Carbon Black (Creative Tools, Halmstad, Sweden), was used to 3D print the sensor parts. The complete 3D printed monolithic unit was printed using a low-cost and open-source FDM 3D printer (FlashForge Inventor, FlashForge Corporation, Jinhua, China). Sensory parts were 3D printed separately due to the fact that the 3D printing transition from one nozzle to another creates printing strings lowering printing quality even though the 3D printer used is equipped with a double nozzle extruder. Rubber adhesive was used to bond the sensory parts to the soft haptic actuator; one part on the bellow and the other sensory part on the head. The authors have not observed any sign of delamination of sensory parts during the experimental tests.

### 2.1. Developing the Soft Haptic System

The computer-aided-design (CAD) models were designed and modelled using Creo Parametric 2.0 (PTC Inc., Boston, MA, USA). The CAD models were sliced using a commercially available slicer known as Simplify3D (Simplify3D LLC, Cincinnati, OH, USA). The printing parameters were adjusted based on recent studies about 3D printing soft pneumatic actuators and sensors using FDM 3D printing [23]. The parameters were adjusted to obtain airtight pneumatic bellow actuators. A positive pressure pneumatic pump (Chicago Air HUSH50) and Festo air-pressure regulators were used to actuate the soft haptic unit(s). The soft haptic units were secured in custom housings 3D printed from PLA using the same 3D printer. The assembly of the soft haptic biofeedback device for balance and proprioception was completed with three haptic actuators integrated with a soft Velcro strap.

### 2.2. Finite Element Modelling

Finite Element modelling was performed on a single actuator to predict its performance in terms of blocked force, stress analysis and deformation. ANSYS Mechanical (ANSYS Workbench 19.1) was used to perform the FEM simulations. Several material models can be utilized to estimate a soft material, TPU Ninjaflex in this case, including linear and non-linear models with the use of experimentally obtained material characterization data in tension and compression, uniaxially and biaxially, planar tension and equibiaxial tension. Among material models, Mooney–Rivlin and Ogden models are the most commonly used model for hyperelastic material modelling. We utilized a 5-parameter Mooney–Rivlin hyperelastic material model for NinjaFlex, and the material model was based on the experimental stress–strain data of NinjaFlex obtained experimentally [24,25] which estimates material behaviour with minimal experimental material data. The assembly was meshed using higher order tetrahedral elements. A Fixed Support boundary condition was imposed on the base of the bellow actuator and a Displacement Support boundary condition of 9.62 mm was imposed on the tip of the sensor. Contact pairs were defined between the different walls that touch upon activation.

The FEM simulations estimated the blocked force of the soft haptic unit accurately. Figure 3 shows the experimental and FEM unloaded states and deformed states for the soft haptic unit. The FEM blocked force obtained is 9.93 N which is 7.02% less than the experimental blocked force obtained of 10.68 N. The main reason for this acceptable difference is that FEM simulations were performed by considering that the soft haptic unit was made of NinjaFlex only. In addition, the geometry of one of the contact surfaces between the actuator and the sensor was simplified by modelling it as a flat surface compared to the elliptical surface in the physical prototype.

### 2.3. Three-Dimensional Printing

The primary objective of this study was to develop a single fully 3D printable soft monolithic haptic unit with conformal capabilities. However, multiple steps were required in manufacturing due to the limitations imposed by using multiple materials (i.e., NinjaFlex and Carbon Black) in 3D printing.

The parts were 3D printed with 100% infill. It is noted that rafts/supports should not be used for printing the parts to avoid any post-processing. Rafts/supports also reduce the quality of the surfaces of the 3D printed parts. Furthermore, the strong bonding that occurs between the main parts and the rafts/supports makes their removal process very challenging. These criteria have been considered during the design of the soft haptic unit.

This problem diminishes the physical properties of the printed parts and affects their effective elastic properties. The main reason for printing the tip separately was to test different tips to ensure that the haptic unit provides the desired comfort levels. The base and the tip were fabricated using NinjaFlex and Carbon black was used for the sensory element. The actuator was activated using positive pressure where the load was transferred via the ring and pillars (Figure 2). This load is then transferred to the tip that produces mechanical stimuli to the wearer.

## 3. Biofeedback Device for Balance Improvement

### 3.1. Developing Pressure Control Unit

In the development of the biofeedback system, a pressure control unit was used to adjust the mechanical stimuli on the thigh. The proposed pressure control unit comprises MATLAB Simulink Real-Time, Humusoft MF644 Data Acquisition Card, Festo air pressure regulators, and Chicago Air HUSH50 air compressor.

Before biofeedback was provided in experiments, the pressure regulation system was calibrated to determine the upper and lower bounds of the CoP of participants. The calibration process is composed of two steps that take 100 s in total. First, the participant is asked to step on the custom force plate and remain static with eyes opened, looking forward, feet together, and hands alongside the body. While the participant keeps this stance for 70 s, the participant’s CoP data are recorded to be used for locating the initial position of the participant’s CoP. The initial position of the CoP is used as a reference when tracking the actual CoP throughout the experiment. Since the initial CoP varies for each subject, it is crucial to repeat this process before initiating the experiment. Recorded data is split into 5 data sets. The initial CoP point is determined by taking the average of these five datasets. The second step is used for locating the expected maximum distances between the initial CoP point and the points that the participant can reach without balance loss. The participant is instructed to lean forward, backward, left, and right with eyes opened, looking forward, feet together, and hands alongside the body. This process takes 30 s. At the end of the second step, maximum recorded CoP values are set to be the furthest points that the participant’s CoP can reach.

After completing the crucial calibration step, biofeedback was provided to participants in the experiments. The actual positions of the CoP of participants were calculated by using a custom-made force plate in experiments. This is explained in the next section in detail. By using the actual position of the CoP and initial calibration, biofeedback was provided through a control algorithm illustrated in Figure 4 to improve the performance of different tasks such as static standing and reaching a location. This process was repeated until the test ended or was terminated by the subject.

### 3.2. Developing Custom Force Plate to Measure CoP

A custom force plate was developed by utilizing Force Sensitive Resistor (FSR) sensors instrumented with a voltage divider to measure their response. The FSRs were strategically placed between two acrylic plates laser cut in the shape of the sole of a human foot, and secured on a 10 mm thick anodized aluminum plate. The centre of pressure of a participant is calculated using Equations (1) and (2) as described in [26].
(1)$$X_{CoP}=\frac{\sum_{i}^{4}C_{i}^{x}F_{i}}{C_{total}^{x}\;F_{total}}$$
(2)$$Y_{CoP}=\frac{\sum_{i}^{4}C_{i}^{y}F_{i}}{C_{total}^{y}\;F_{total}}$$

In these equations, $X_{COP}$ and $Y_{COP}$ represent the projection coordinates of the participant’s CoP on the force plate’s planar surface. The medial–lateral direction is represented as $X_{COP}$ coordinate and the anterior–posterior direction is represented as $Y_{COP}$ coordinate of the CoP. $C_{total}^{x}$ and $C_{total}^{y}$ represent the distances between the force resistors placed underneath the acrylic plates. $C_{i}^{x}$ and $C_{i}^{y}$ represent the instantaneous distances that are calculated in reference to the CoP. While $F_{i}$ represents the instantaneous force values measured by the force resistors, $F_{total}$ is the sum of the measured forces.

## 4. Experiments

### 4.1. Experimental Procedure

To validate the performance of the proposed biofeedback device, we conducted experiments with 5 subjects. One subject is illustrated standing on the force plate with the soft haptic biofeedback device system strapped around his tibia in Figure 5. Participants were aged 18 years old and over, had no known orthopaedic or neurological deficits and/or balance disorders, were able to maintain static standing and/or sitting balance, and were able to communicate clearly. All subjects gave their informed consent for inclusion before they participated in the study. The study was conducted in accordance with the National Statement on Ethical Conduct in Research Involving Humans, and the protocol was approved by the Human Research Ethics Committee at the University of Wollongong (HREC Ethics Number 2019/423).

This study involved two major parts:

**Part A:** The pressure threshold of the soft haptic biofeedback system was tested on each subject. This was undertaken using the soft haptic biofeedback system during static standing, with eyes opened, looking forward, feet together, and hands alongside the body, and pressure input was gradually increased. This part of the study provides the minimum perceived pressure sensation and maximum pressure level that is comfortable for the subject. This was then used to determine the pressure limits to ensure the device did not cause discomfort to the subject. If this approach did not eliminate discomfort, the experiment was to be terminated.

Balance sensors were also calibrated in this part of the study, asking the subject to perform various movements (i.e., perform body sway movements in the direction indicated by activation of each actuator) while static standing, with eyes opened, looking forward, feet together, and hands alongside the body and the same exercise with eyes closed.

The soft haptic biofeedback device was strapped to the subject. The location of the device was determined based on the subject’s perception; large muscular areas such as thigh muscles might be difficult to feel pressure input which would require higher pressure levels, on the other hand, lean muscular areas such as the forearm or tibia may provide better sensation. This was determined during the calibration phase of the study. Then, the subject was given a three (3) minute practice period to get familiar with the soft haptic biofeedback system; for instance, the subject was asked to tilt their body toward left/right and forward/backward to observe how the soft haptic biofeedback system provides an amplitude of pressure feedback onto their skin depending on the amount of tilting of their body.

**Part B:** On completion of Part A, the subject proceeded with the performance of actual tests of the soft haptic biofeedback system for balance:

Test 1: The subject remained static standing, with eyes opened, looking forward, feet together, and hands alongside the body with no biofeedback. The subject stood in this position for 5 min.

Test 2: The subject remained static standing, with eyes closed, looking forward, feet together, and hands alongside the body with no biofeedback. The subject stood in this position for 5 min.

Test 3: The subject remained static standing, with eyes closed, looking forward, feet together, and hands alongside the body with biofeedback. The subject stood in this position for 5 min.

Test 4: The subject remained static standing, with eyes opened, looking forward, feet together, and hands alongside the body with biofeedback. The subject stood in this position for 5 min.

Test 5: The subject remained static standing, with eyes opened, looking forward, feet together, and hands alongside the body with no biofeedback. Then, the subject was asked to reach a target location/object location fixed by leaning their trunk as illustrated in Figure 5. The subject repeated this task 3 times. The target/object was removed and the subject was instructed to reach the same target location/object location by leaning their trunk.

Test 6: The subject remained static standing, with eyes opened, looking forward, feet together, and hands alongside the body with biofeedback. Then, the subject was asked to reach a target location/object location fixed by leaning their trunk. The subject repeated this task 3 times. The target/object was removed and the subject was asked to reach the same target location/object location by leaning their trunk with the presence of biofeedback provided depending on the amount of tilting of their body.

### 4.2. Experimental Results

The static standing experiments as the first part of the pilot study, i.e., Test 1, Test 2, Test 3 and Test 4. Figure 6 illustrates the displacement of the CoP of a subject when biofeedback was not provided and the subject was asked to remain steady by keeping their CoP fixed (i.e., within the blue circle shown in Figure 5 and Figure 6) for 5-min with closed eyes. While the blue circle represents the desired area for the CoP, the orange curve represents the actual position of the CoP of the subject. As shown in these figures, we observed large CoP movements and the subject could not keep their CoP within the desired area most of the time during the experiment. The subject was not aware of this large CoP movement due to the lack of biofeedback.

Figure 7 illustrates the same experiment when biofeedback was provided to the subject. While the black curve represents the actual position of the CoP within the desired area, the red, green, brown, purple, and blue curves represent the actual CoP position outside the desired circular area where the biofeedback system was activated. As shown in this figure, the subject could immediately respond and move their CoP into the desired area using the proposed biofeedback system. The CoP of the subject stayed inside the desired area most of the time during the experiment.

To verify the performance of the biofeedback system, we repeated this experiment with 5 subjects. The results of these experiments are outlined in Figure 8. The success rate is calculated by considering the time over which the CoP of a subject stayed within the desired circular area as a proportion of the total trial time. As shown in Figure 8, the proposed biofeedback system improved the participants’ ability to remain within the target circle about the CoP, thus participants’ balance was improved with the provided biofeedback.

The second major part of the experiments assesses the subject’s ability to recall their body position by employing a target object to reach by leaning forward, i.e., Test 5 and Test 6. This is assessed by utilizing the CoP data collected during Test 5 and Test 6. It is noted that the desired location of the CoP is different for each subject due to biomechanical differences such as height and body mass index. Figure 9 illustrates the displacements of the CoP of two subjects along the *Y*-axis when they attempted to reach the target object location by leaning forward. Despite some differences, this figure clearly shows that the proposed biofeedback system allows subjects to locate their CoP to the desired location.

We repeated the reaching a desired location/object location experiment with 5 subjects. The success rate was calculated by considering the time in which the CoP of a subject stayed within the target object location as a proportion of the total trial time. As illustrated in Figure 10, the success rate of reaching the target location was higher when biofeedback was provided to the subjects.

### 4.3. Discussion

This study is organised to conduct a pilot investigation on the effects of soft haptic biofeedback devices on the postural balance control and proprioception of the wearer. The pilot tests were conducted with participants aged 18 years old and over, had no known orthopaedic or neurological deficits and/or balance disorders, were able to maintain static standing and/or sitting balance, and were able to communicate clearly. Before biofeedback was provided in the experiments, the pressure regulation system was calibrated to determine the upper and lower bounds of the CoP of participants. The calibration process is composed of two steps that take 100 s. First, the participant is asked to step on the custom force plate and remain static with eyes opened, looking forward, feet together, and hands alongside the body. While the participant keeps this stance for 70 s, the participants’ CoP data are recorded to locate the initial position of the participant’s CoP. The initial position of the CoP is used as a reference when tracking the actual CoP throughout the experiment. Since the initial CoP varies for each subject, it is crucial to repeat this process before initiating the experiment. Recorded data are split into five datasets. The initial CoP point is determined by taking the average of these five datasets. The second step is used for locating the expected maximum distances between the initial CoP point and the points that the participant can reach without balance loss. The participant is instructed to lean forward, backward, left, and right with eyes opened, looking forward, feet together, and hands alongside the body. This process takes 30 s. At the end of the second step, maximum recorded CoP values are set to be the furthest points that the participant’s CoP can reach.

After completing the crucial calibration step, biofeedback was provided to participants in the experiments. The actual positions of the CoP of participants were calculated by using a custom-made force plate in experiments. This is explained in the next section in detail. Using the actual position of the CoP and initial calibration, biofeedback was provided through a control algorithm illustrated in Figure 4 to improve the performance of different tasks such as static standing and reaching a location. This process was repeated until the test ended or was terminated by the subject.

The ultimate intention of the soft haptic biofeedback device is to employ in more thorough human–robot/computer interactions, such as rehabilitation activities for individuals who have limited proprioception and poor balance control, such as individuals with sensorimotor dysfunction. In this study, we have demonstrated that a soft haptic biofeedback device, actuated with positively pressurized bellow-like soft chambers, is safe and highly compatible in utilization within human-computer interactions, effective in reducing CoP deviation during standing and voluntary movement, and substituting (when the wearer’s eyes closed)/augmenting (when the wearer’s eyes opened) proprioceptive feedback.

In Part A of this pilot study, experiments were focused on maintaining balance, operationalised as centre of pressure (CoP), during steady-standing. Results presented in Figure 6 and Figure 7 clearly indicate that when participants closed their eyes, they showed higher performance (smaller CoP deviation) if they were assisted by the soft biofeedback to remain in their target CoP area compared to their performance when the biofeedback was not present. We also demonstrated that even when visual information was available (i.e., the wearer’s eyes being open), wearers’ performance was lower than their performance with the soft biofeedback during eyes closed tests, as illustrated in Figure 8. This result is perhaps surprising since visual feedback is the dominant form of sensory feedback by which humans guide their movements, although the lack of visual cues related to the target CoP region in this study likely explains the result. It does, however, also emphasise the effectiveness of providing a source of information related to the CoP position for balance regulation, as this information must usually be calculated by the central nervous system indirectly.

In Part B of the pilot study, participants were asked to reach a target body position by leaning their trunk as illustrated in Figure 5 to investigate proprioceptive feedback. While each subject repeated this task three times, the target/object was removed and they were inquired to reach the same target object location by leaning forward. As per data collection, the subject was not informed how to lean their body, for instance, either move their trunk or lean from their ankles. With the presence of the soft biofeedback monitored by their CoP position, participants were able to move to the target location more quickly and accurately compared to their attempts with no biofeedback provided—this was reflected by higher success rates during movements with biofeedback as shown in Figure 10.

## 5. Conclusions

In this paper, a pilot study of a novel soft haptic biofeedback device was conducted to assess its effectiveness in improving the balance and proprioception of the wearer through a non-invasive soft biofeedback stimulus. The device was developed according to the principles of soft robotics with the use of soft bellow-type actuators, actuated with positive pressure to create mechanical stimulation of sensory receptors in the skin of the wearer. The pilot investigation of the effects of soft haptic biofeedback on substitution and/or augmentation of sensory functions associated with postural balance and proprioception of the wearer is outlined in which centre of pressure excursions during standing and voluntary body movements were assessed in five neurologically healthy participants. The results from these experimental studies show that participants were performed in higher success rates in maintaining their balance and recalling their body posture more accurately to a target when soft biofeedback is present which was mapped with corresponding CoP axes.

Future studies will focus on testing our soft haptic biofeedback device within thorough investigations of a larger group of healthy participants, a wider spectrum of participation criteria with the inclusion of subjects with limited sensory functions, and in the context of more diverse movements. Further, the authors will extend this study by intergrating with a robot-assisted rehabilitation device to complement motor and sensory function loss within our undergoing projects.

## Figures and Tables

**Figure 1 sensors-22-03779-f001:**
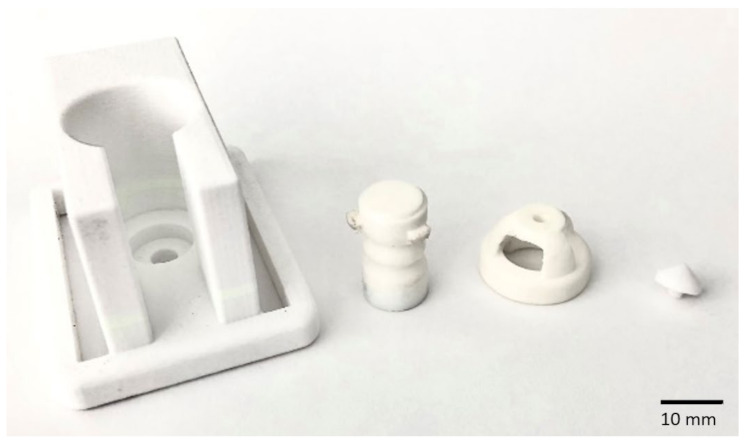
The 3D printed Soft Haptic Biofeedback system: from left, custom housing, the soft bellow pneumatic actuator, the compliant head and an interchangeable tip.

**Figure 2 sensors-22-03779-f002:**
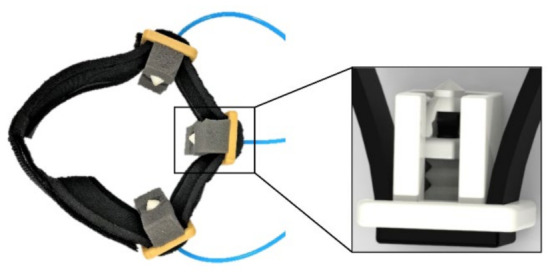
Soft haptic biofeedback system with three units with a padded Velcro strap.

**Figure 3 sensors-22-03779-f003:**
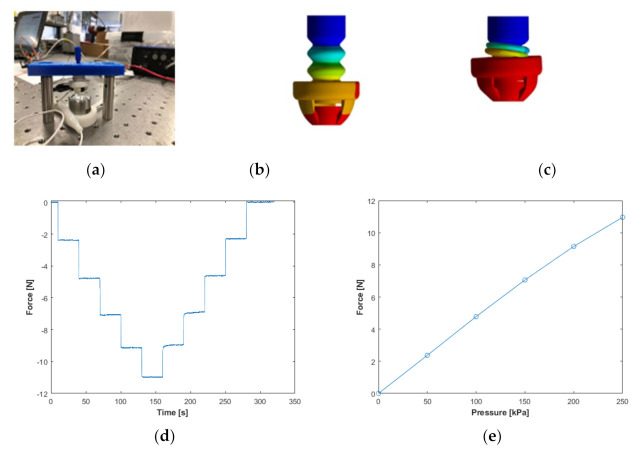
Blocked force measurement experimental setup and FEA comparison: (**a**) haptic unit measuring blocked force with 6−axis force sensor, (**b**) FEM simulations in default state, (**c**) deformed state when a 9.62 mm displacement input is applied, (**d**) the soft haptic biofeedback device was subjected to variable pressure values of 50, 100, 150, 200 and 250kPa where the measured blocked forces in the transient, and (**e**) steady-state mode.

**Figure 4 sensors-22-03779-f004:**
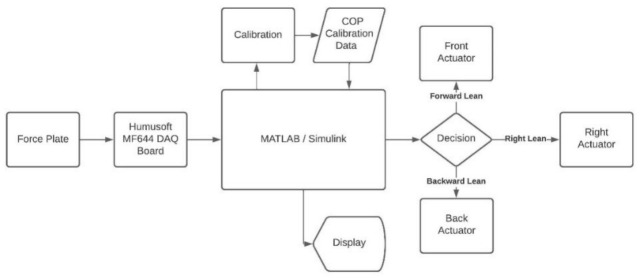
A flowchart that represents the control algorithm of the proposed biofeedback system.

**Figure 5 sensors-22-03779-f005:**
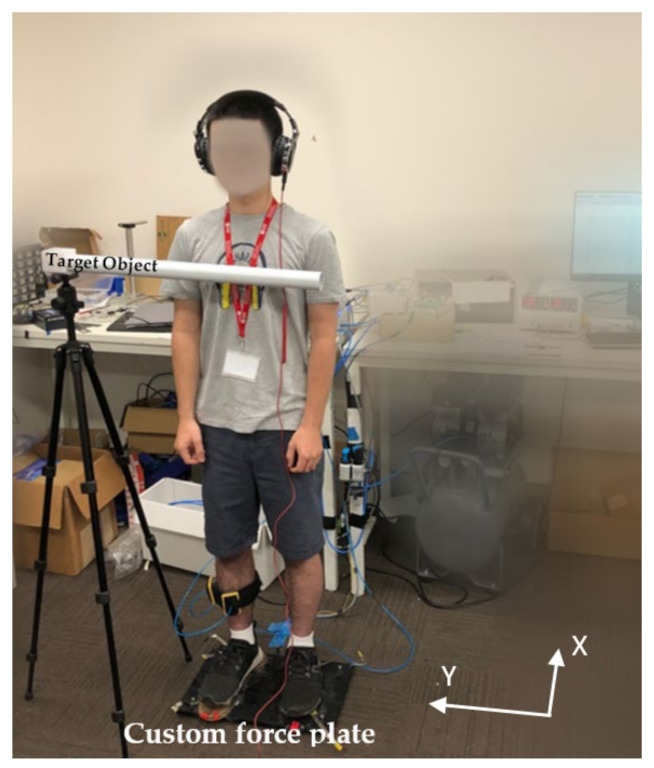
Participant on the custom-built force plate.

**Figure 6 sensors-22-03779-f006:**
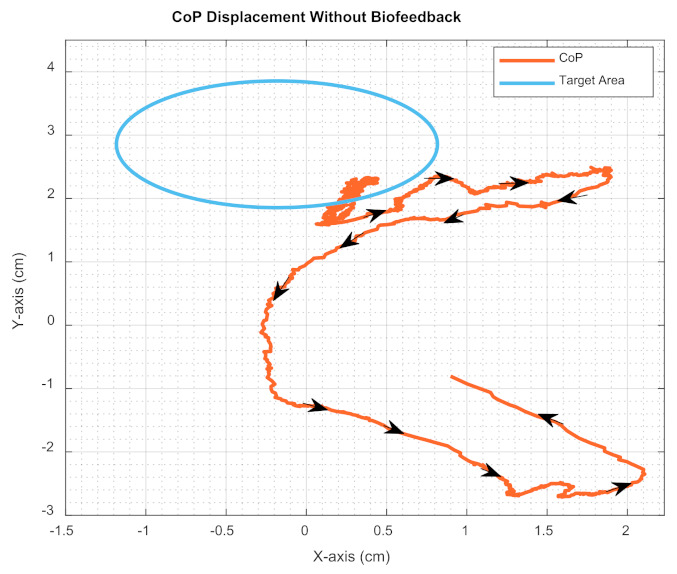
CoP displacement without biofeedback.

**Figure 7 sensors-22-03779-f007:**
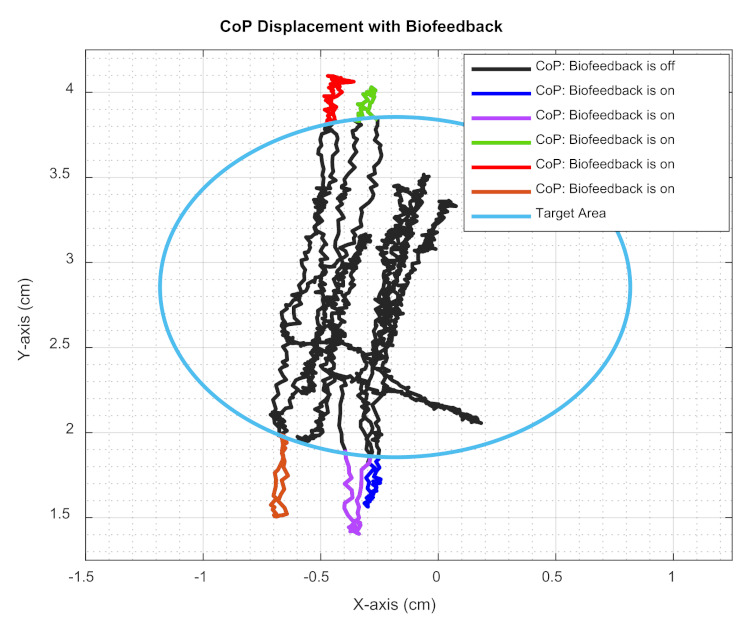
CoP displacement with biofeedback.

**Figure 8 sensors-22-03779-f008:**
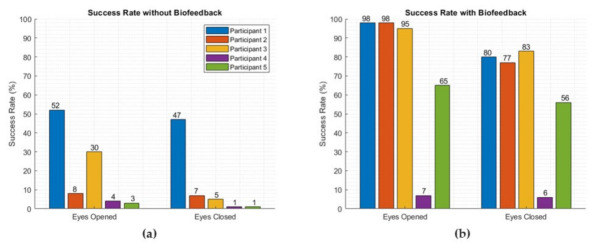
The success rate of static standing experiments with five subjects: (**a**) without soft haptic biofeedback and (**b**) with soft haptic biofeedback presence.

**Figure 9 sensors-22-03779-f009:**
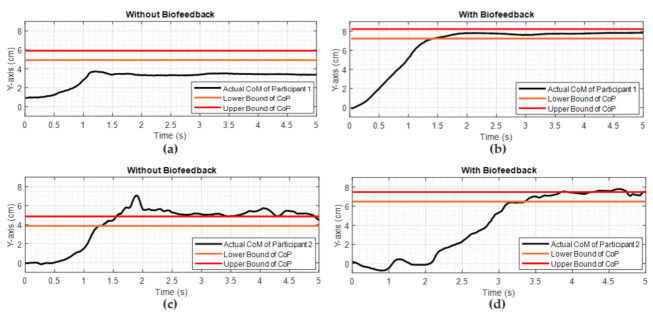
Reaching a desired CoP location experiments of two subjects: Participant 1 when received (**a**) no soft haptic biofeedback, (**b**) with biofeedback, and Participant 2 when received (**c**) no soft haptic biofeedback, (**d**) with biofeedback.

**Figure 10 sensors-22-03779-f010:**
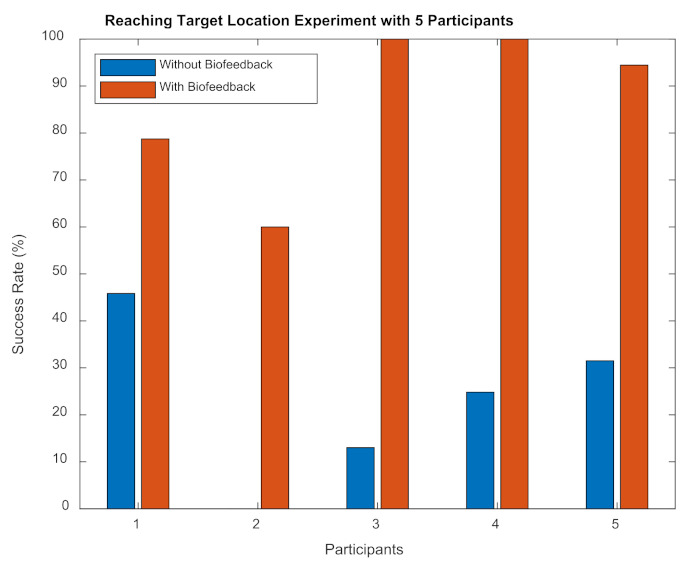
The success rate of reaching a target location experiment with 5 subjects.

## Data Availability

The data presented in this study are available on request from the corresponding authors.

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
