# Peer review of "Novel Soft Haptic Biofeedback—Pilot Study on Postural Balance and Proprioceptionâ€"

_sensors, 2022, doi:10.3390/s22103779_

Round 1
Reviewer 1 Report
In this article, a soft monolithic haptic biofeedback device is introduced that has been developed through fused deposition modeling (FDM) 3D printer using Polyurethane. A validation study has been performed on 5 subjects to explore the effectiveness of the device. The results reveals the promising performance of the device, however, there are a few major requirements for this study to get accomplished:
- How the accuracy of stress analysis performed performed via FEM simulation is calculated? It is highly suggested that the authors study the performance of their final device (rather than using NinjaFlex data) under physical compression (uniaxial tensile testing) and validate their simulation results.
- In the fabrication section, it is not clear where the sensory element is located in the device and how it has been integrated into the TPU body structure? Are they printed together? If not, how carbon black part is attached to TPU without delamination.
- The experiments are designed in two modes of static-standing and leaning in the direction of the target location. It would be nice to see what are the effects of motion artifacts on the device performance if the experiments includes a variety of movements in different directions rather than just one.
- How would the device perform if the body sway in a direction different from the one indicated by the activation of each actuator?
- The results revealed that visual information (eyes being open) has almost no positive effect on the success rate, with or without using the biofeedback. This cannot be attributed to the performance of the device. More accurately designed experiments is suggested to explore why this was observed.
- The user study has been performed only on 5 subjects and that is when one of them (participant 4) does not show any positive success rate% to the biofeedback device. It is highly suggested that the authors perform the experiments with a larger group of subjects specifically from a variety of ages including elderly subjects as they are the main target group and the ultimate market and probably will impose unintentional swayings that would help to evaluate the real performance of the device.
Author Response
The authors kindly appreciate for the reviewer's time, effort and constructive feedback to heighten the quality of the paper. The response to the reviewer's comments are combined with the manuscript revised in the same file and attached below.

Reviewer 2 Report
The current study proposes a soft haptic interface as part of a biofeedback system to support balance. The paper is well motivated, the methodology for designing the soft haptic interface is well articulated, and the experimental study is sufficiently described. Whereas the merits of the work are clear, there are several comments that may help improve the quality of this work.
1. The most important aspect is that this device is used for biofeedback (the human subject experiment is with healthy subjects). The introduction refers to stroke patients. I suggest the Authors to make a clear statement whether this is a biofeedback device or a medical one? In case the intention is to use it for patients, the human subject experiment would have to be conducted with the target group (patients). Please clarify.
2. The calibration process is not clearly described. How the device is calibration for the subject’s CoP? Please elaborate.
3. The Authors state “Modular compliant tips on the head of the unit were designed with various shapes to 107 assess the comfort levels of subjects when the bellow-actuator is activated using a positive 108 pressure to push the tip of the unit against the skin of the user.”. However, the paper does not include any details about the specific tip shapes/configurations and how they were evaluated. A brief description would properly motive the use of the current tip.
4. Maybe this is obvious to the Authors (but not necessarily to the reader): why using three actuators to signal back, front, and right. Is there any need to provide feedback for the left direction? Do you assume that the user uses two of these interfaces (left/right legs)? Please clarify.
5. Figures 8 and 10 show extremely high inter-subject variability in performance. For example, some of the subjects had zero success rate. Can you shed more light into the reasons behind this high variability in performance?
6. It is not clear why the Authors are comparing the performance when biofeedback is turned on or off. It is clear that participants would perform better when receiving additional guidance no matter what modality. It would be interesting to compare the performance when haptic and visual biofeedback are utilized? Why do we need this wearable soft haptic interface, does it perform any better compared to a simple visual biofeedback system?
7. It would be useful to add a size scale to Figure 2 to provide a perspective on the size of the soft actuator.
8. There are a few typos that need to be correct. I suggest proofreading the paper by an English native speaker.
Line 62: such as such as
Line 242: wee
Line 308: is asssed by
Author Response

(The authors gave the same response as above.)

Round 2
Reviewer 1 Report
The authors have almost addressed all my comments. I do not have any additional comments.